# Targeting SARS-CoV-2 receptor-binding domain to cells expressing CD40 improves protection to infection in convalescent macaques

Romain Marlin [1,15], Veronique Godot[2,3,15], Sylvain Cardinaud[2,3,15], Mathilde Galhaut[1], Severin Coleon[2,3], Sandra Zurawski[2,4], Nathalie Dereuddre-Bosquet [1], Mariangela Cavarelli [1], Anne-Sophie Gallouët[1], Pauline Maisonnasse [1], Léa Dupaty[2,3], Craig Fenwick[5,6], Thibaut Naninck [1], Julien Lemaitre[1], Mario Gomez-Pacheco[1], Nidhal Kahlaoui[1], Vanessa Contreras[1], Francis Relouzat[1], Raphaël Ho Tsong Fang[1], Zhiqing Wang[2,4], Jerome Ellis III [2,4], Catherine Chapon[1], Mireille Centlivre [2,3], Aurelie Wiedemann [2,3], Christine Lacabaratz [2,3], Mathieu Surenaud[2,3], Inga Szurgot[7], Peter Liljeström[7], Delphine Planas[2,8,9], Timothée Bruel [2,8,9], Olivier Schwartz [2,8,9], Sylvie van der Werf [10,11], Giuseppe Pantaleo [2,5,6], Mélanie Prague[2,12,13], Rodolphe Thiébaut [2,12,13], Gerard Zurawski[2,4], Yves Lévy [2,3,14,15✉] & Roger Le Grand [1,15✉]

Achieving sufficient worldwide vaccination coverage against SARS-CoV-2 will require additional approaches to currently approved viral vector and mRNA vaccines. Subunit vaccines may have distinct advantages when immunizing vulnerable individuals, children and pregnant women. Here, we present a new generation of subunit vaccines targeting viral antigens to CD40-expressing antigen-presenting cells. We demonstrate that targeting the receptor-binding domain (RBD) of the SARS-CoV-2 spike protein to CD40 (αCD40.RBD) induces significant levels of specific T and B cells, with long-term memory phenotypes, in a humanized mouse model. Additionally, we demonstrate that a single dose of the αCD40.RBD vaccine, injected without adjuvant, is sufficient to boost a rapid increase in neutralizing antibodies in convalescent non-human primates (NHPs) exposed six months previously to SARS-CoV-2. Vaccine-elicited antibodies cross-neutralize different SARS-CoV-2 variants, including D614G, B1.1.7 and to a lesser extent B1.351. Such vaccination significantly improves protection against a new high-dose virulent challenge versus that in non-vaccinated convalescent animals.

[1] Center for Immunology of Viral, Auto-immune, Hematological and Bacterial diseases (IMVA-HB/IDMIT), Université Paris-Saclay, Inserm, CEA, Fontenay-aux-Roses, France. [2] Vaccine Research Institute, Creteil, France. [3] Inserm U955, Créteil, France. [4] Baylor Scott and White Research Institute and INSERM U955, Dallas, TX, USA. [5] Service of Immunology and Allergy Lausanne University Hospital, Lausanne, Switzerland. [6] Swiss Vaccine Research Institute, Lausanne University Hospital, University of Lausanne, Lausanne, Switzerland. [7] Department of Microbiology, Tumor and Cell Biology, Karolinska Institutet, Stockholm, Sweden. [8] Virus & Immunity Unit, Department of Virology, Institut Pasteur, Paris, France. [9] CNRS UMR 3569, Paris, France. [10] Molecular Genetics of RNA Viruses, Department of Virology, Institut Pasteur, CNRS UMR 3569, Université de Paris, Paris, France. [11] National Reference Center for Respiratory Viruses, Institut Pasteur, Paris, France. [12] Univ. Bordeaux, Department of Public Health, Inserm Bordeaux Population Health Research Centre, Inria SISTM, Bordeaux, France. [13] CHU Bordeaux, Department of Medical information, Bordeaux, France. [14] AP-HP, Hôpital Henri-Mondor Albert-Chenevier, Service d'Immunologie Clinique et Maladies Infectieuses, Créteil, France. [15] These authors contributed equally: Romain Marlin, Veronique Godot, Sylvain Cardinaud, Yves Lévy and Roger Le Grand. ✉email: yves.levy@inserm.fr; roger.le-grand@cea.fr

Coronavirus-induced disease 2019 (COVID-19) is caused by a zoonotic virus, severe acute respiratory syndrome coronavirus 2 (SARS-CoV-2), which has rapidly spread during the last year and a half, infecting over 100 million humans and causing more than two million deaths worldwide. Durable control of the pandemic requires mass vaccination strategies, for which the first vaccine candidates became available at the end of 2020. Although there are a limited number of previously licensed vector-based vaccines for human use, recombinant DNA vector and synthetic mRNA vaccines have nevertheless become the most advanced in the fight against COVID-19 because of the many possibilities offered for genetic engineering and rapid scalability[1–4]. Given that the benefits outweigh the risks for their use in humans, several vaccines, including mRNA-derived, vector-based vaccines and virus inactivated vaccines have been authorized for an emergency use to fight the spread of the disease in humans[3,5–11]. The estimated efficacy after phase III clinical trials and first efficacy assessment during vaccination campaigns was approximately 60–95% in preventing COVID-19. Long-term efficacy data will be critical for estimating their impact on pro-gression of the pandemic. Initial reports on adverse events may not limit their deployment, but safety assessments require extended follow-up. Further evaluations are still be needed to assess the efficacy of the vaccines against next SARS-CoV-2 variants and in preventing asymptomatic infections and reducing viral shedding to the level required to prevent secondary transmission[12–15]. If not efficiently prevented, asymptomatic infections in combination with reduced mask wearing and social distancing could result in significant continuing circulation of the virus[5].

A new generation of COVID-19 vaccines is needed to coun-teract the development of the pandemic. Providing the necessary billions of doses to achieve sufficient global coverage will not be possible with any single product. In addition, there are uncer-tainties about the long-term efficacy and safety of these first-in-class vector or mRNA vaccine platforms, with a limited history of use, particularly in vulnerable individuals, including frail, older individuals, people with co-morbidities, and immunosuppressed patients. Importantly, the use of vector-based vaccines will require cautious and long-term safety assessment when using in children and pregnant women. Although younger individuals are less prone to develop severe disease, they are susceptible to mild COVID-19 or asymptomatic infection and may facilitate circu-lation of the virus and the potential for further mutation. Control of the pandemic will also require the mass immunization of children.

The constraints of antigen design and engineering and the time required for the production of large numbers of doses make subunit vaccines difficult to develop as first countermeasures for suddenly emerging and non-anticipated epidemics. However, licensed subunit vaccines have proven tolerability and safety in diverse population classes[16]. Several adjuvanted SARS-CoV-2 spike protein vaccines are able to elicit neutralizing antibodies to protective levels in relevant animal models, including non-human primate (NHP) challenge studies[17–19]. These advantages may be decisive in the development of the next-generation vaccines aimed at controlling the long-term circulation of SARS-CoV-2, in particular if the virus continues provoking seasonal epidemic waves of COVID-19.

Dendritic cells (DCs) are immune system controllers that can deliver differential signals to other immune cells through inter-cellular interactions and soluble factors, resulting in a variety of host immune responses of varying quality. Targeting vaccine antigens to DCs via surface receptors represents an appealing strategy to improve subunit-vaccine efficacy while reducing the amount of required antigen. Direct delivery of the antigen, which can additionally activate cell receptors, may also evoke a danger signal, stimulating an immune response without the need for additional immune stimulants, such as adjuvants. Among the various DC receptors tested, including lectins and scavenger receptors, we reported the capacity of vaccines targeting diverse viral antigens to CD40 expressing antigen-presenting cells in evoking strong antigen-specific T- and B-cell responses[20–24]. We confirmed the advantage of CD40 targeting over the not targeted immunogen control, in a preliminary study of cynomolgus macaques immunization with HIV envelope gp140 glycoprotein (Supplementary Fig. 10).

In this work, we develop a vaccine that targets the receptor-binding domain (RBD) of the SARS-CoV-2 spike antigen to the CD40 receptor (αCD40.RBD). We prove its immunogenicity in two different animal models. A single dose of the αCD40.RBD administered without adjuvant boosts the protective response in COVID-19 convalescent NHPs.

## Results

**The αCD40.RBD vaccine targets and activates antigen pre-senting cells.** The human ACE2 receptor is the crucial target for the receptor-binding domain (RBD) of the spike (S) protein of SARS-CoV2, for which this strong interactive synapse assists viral entry into host cells[25]. The RBD is a logical target for the development of neutralizing antibodies, as well as serving as a potential source of T-cell epitopes to elicit cellular immune responses. Thus, we engineered vectors expressing SARS-CoV-2 RBD (residues 318–541 of sequence ID: YP_009724390.1) fused to the C-termini of the anti-human CD40 humanized 12E12 IgG4 antibody[24,26,27] to generate the αCD40.RBD vaccine (Fig. 1a).

As evaluated by a solid-phase direct-binding assay[24], there was no significant difference in CD40 binding affinity (EC50 30 pM) between the 12E12 anti-CD40 monoclonal antibody and 12E12 anti-CD40 fused to RBD (EC50 35 pM) (Fig. 1b and Supplementary Fig. 1a). We have previously shown that 12E12 anti-CD40 fused to viral antigens, like influenza and HIV, enhances CD40-mediated internalization and antigen-presentation by mononuclear cells and ex vivo generated monocyte-derived DCs[20,26]. Similarly, we show here that the αCD40.RBD vaccine binds (Fig. 1c; Supplementary Fig. 1b, c) and activates (Fig. 1d; Supplementary Fig. 1d) macaque monocytes, DCs, and B cells obtained from peripheral blood mononuclear cells (PBMCs).

**The αCD40-RBD vaccine induces human B- and T-cell responses in humanized mice.** We first assessed the immuno-genicity of the αCD40.RBD vaccine in NSG (NOD/SCID γc$^{-/-}$) mice with a human immune system (hu-mice) generated by reconstituting newborns with human fetal liver hematopoietic stem cells (Fig. 1e). A single injection of αCD40.RBD (10 μg), adjuvanted with polyinosinic-polycytidylic acid (Poly-IC, 50 μg), by the intraperitoneal route was sufficient to elicit SARS-CoV-2 S protein-specific IgG-switched human B cells in the blood of 50% of immunized mice (Fig. 1f). At week 6, one week after the last αCD40.RBD boost, unbiased t-SNE analysis of the splenic human CD19$^+$ B cells revealed cell clusters corresponding to well-described subsets of terminally differentiated plasma cells (PCs), early plasma blasts (PBs), and a contingent of PBs and immature PCs in the vaccine groups but not controls (Fig. 1g–j). At the same time point, splenic SARS-CoV-2 S protein-specific IgG-switched human B cells were detected in all vaccinated hu-mice (Fig. 1i), mainly of the PB and immature PC phenotype (Fig. 1i). All spike protein-specific IgG-switched human B cells expressed CXCR4 and a discrete cell island was observed in the t-SNE analysis driven by high expression of CCR10 (Fig. 1k), which was

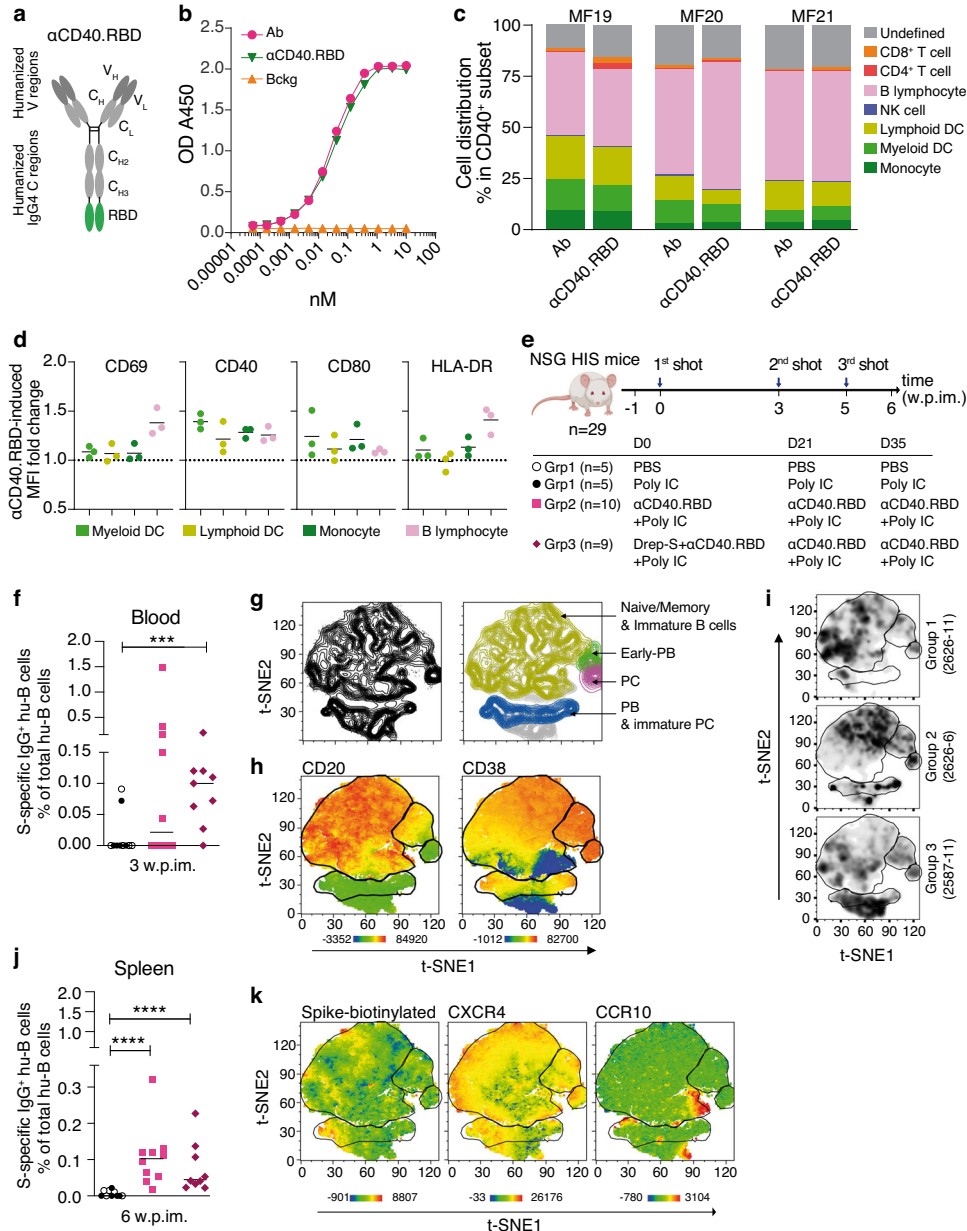

**Fig. 1 αCD40.RBD vaccine targeting and immunogenicity in hu-mice. a** Picture of αCD40.RBD vaccine construct. **b** Binding to solid-phase attached human CD40 ectodomain protein by anti-CD40 12E12 human monoclonal antibody (Ab, filled pink circles) by the anti-CD40 12E12-RBD vaccine (αCD40.RBD, filled green triangles) and control IgG4 (bckg, filled orange triangles). **c** Binding of 12E12 antibody (Ab) and αCD40.RBD vaccine to CD40-expressing PBMCs of three naive cynomolgus macaques measured by flow cytometry. Cell subsets were defined by the gating strategy shown in Supplementary Fig. 1b. **d** Fold change of the geometric mean fluorescence intensity (MFI) of activation markers after 18 h of incubating NHP ($n = 3$) PBMCs with the αCD40.RBD vaccine for cell subsets targeted by the αCD40.RBD vaccine and identified in (**c**). Mean value is indicated by horizontal bar. **e** Schematic overview of vaccination strategies in NSG humanized (hu) mice, including three experimental groups, 9 to 10 animals/group. **f** SARS-CoV-2 S protein-specific IgG-switched human B-cell frequencies within the hu-B cells in the blood of hu-mice three weeks after the priming injection. Individual values are presented, along with the median. **g** Flow cytometry t-SNE analysis of splenic CD19 + B cells based on five markers (mCD45, hCD45, hCD19, hCD20, hCD38) showing the clustering of PCs, early plasma blasts (PBs), and a population of PBs and immature PCs. Merged t-SNE plots for all samples ($n = 29$). **h** Mapping of CD20 and CD38 onto the splenic hu-B-cell clusters obtained following t-SNE analysis. Merged t-SNE plots for all samples ($n = 29$). **i** Representative examples of t-SNE of one hu-mouse from each group. **j** SARS-CoV-2 S protein-specific IgG-switched human B-cell frequencies within the hu-B cells in the spleen of hu-mice six weeks after the priming injection. Individual values are presented, along with the median. **k** Mapping of the SARS-CoV-2 S protein trimer, CXCR4, and CCR10 onto the splenic hu-B-cell clusters obtained following t-SNE analysis. **f–j** Mann-Whitney U-tests were used for comparisons. ***$p < 0.001$, ****$p < 0.0001$.

confirmed using manual back gating (Supplementary Fig. 2a). We next evaluated the capacity of the vaccines to induce specific and functional CD4$^+$ and CD8$^+$ memory T cells. The Th1 (IFN-γ$^{+/-}$ IL-2$^{+/-}$ TNF-α) type CD4$^+$ T-cell responses and IFNγ-secreting CD8$^+$ T-cells were observed for the vaccinated hu-mice following ex vivo stimulation of splenocytes with RBD peptide pools (Supplementary Fig. 2d, e). We confirmed the presence of human CD8$^+$ T cells specific for the predicted optimal epitopes from SARS-CoV-2 RBD protein in the spleens of vaccinated hu-mice using HLA-I tetramers (Supplementary Fig. 2f, g).

Subunit vaccines could also be considered as boosters for other type of vaccines in human vaccination campaigns. Thus, in addition to a homologous prime-boost regimen, we tested the capacity of αCD40.RBD to boost heterologous priming with a vector-based vaccine. The DNA-launched self-amplifying RNA replicon vector encoding the SARS-CoV-2 spike glycoprotein (DREP)-S is a previously described platform[28] based on the alphavirus genome encoding the genes for the viral RNA replicase but lacking those encoding the structural proteins of the virus[29]. We demonstrated that in the two vaccinated groups, the prime boost strategy containing αCD40.RBD efficiently elicited B- and T-cell SARS-CoV-2 specific responses (Fig. 1f, j; Supplementary Fig. 2d, e). In both vaccinated groups, we showed an expansion of effector memory CD4 and CD8$^+$ T cells (CD45RA$^-$CD27$^-$) (Supplementary Fig. 2b, c).

**The αCD40.RBD vaccine recalls specific immune responses in convalescent macaques**. The immunogenicity observed in the hu-mice model are consistent with those of our previous CD40-targeted influenza and HIV vaccine studies[21,22,26,27] and demonstrate that αCD40.RBD could be a potent prime or boost vaccine for eliciting RBD-specific T- and B-cell responses[19]. Our preliminary data using the HIV envelope glycoprotein as antigen demonstrated the value of CD40 targeting in NHP (Supplementary Fig. 10) over the use of non-targeted antigen. The two anti-CD40 vaccines, bearing with HIV envelope or the SARS-CoV-2 RBD, have been constructed using identical strategies to anchor the viral antigens. Both vaccines have one monomeric HIV Env or RBD fused to each heavy chain forming the Fc domain of the IgG4. In total each construct has two monomers of Env or RBD similarly attached to the anti-CD40 platform. We, therefore, surmise that both constructs should have similar capabilities in targeting the antigen presenting cells expressing human CD40. The effect on CD40 expressing cells of the αCD40.RBD vaccine is also confirmed in Fig. 1. In addition, we previously showed that nanomolar amounts of αCD40 HIV vaccine can elicit in vitro recall responses in PBMCs collected from individuals primed by the natural viral infection[26]. Altogether, the immunogenicity studies in hu-mice, our previous data and the preliminary results with the HIV vaccine, encourage us to test the hypothesis that the αCD40.RBD vaccine can efficiently elicit recall responses in vivo in SARS-CoV-2 convalescent individuals. The improved immunogenicity obtained by CD40 targeting with our HIV vaccine in NHP (Supplementary Fig. 10) and the stimulating capacity of the αCD40.RBD vaccine (Fig. 1) also suggested that adjuvant may not be necessary to elicit a protective recall response in SARS-CoV-2 convalescent individuals. We thus subcutaneously injected six convalescent cynomolgus macaques with 200 μg of the vaccine without adjuvant. An additional 12 animals (six convalescent and six naive) were injected with PBS as controls (Fig. 2a). All the convalescent macaques, randomly distributed between the vaccine and control groups, had been infected approximately six months before (range = 26–24 weeks) with SARS-CoV-2 in a study to evaluate pre-exposure or post-exposure prophylaxis with hydroxychloroquine (HCQ). No evidence of antiviral efficacy[30] of

HCQ was observed and after this first exposure to the virus, all animals developed similar profiles of viral load (Supplementary Fig. 3a and b) and suffered from transient and moderate disease, resulting in increased levels of anti-S IgG antibodies detected in the serum (Fig. 2b). At the time of the αCD40.RBD-vaccine injection, anti-S IgG levels in the two groups of convalescent macaques were comparable and in the average range of specific responses detected in the sera of convalescent patients (Fig. 2c). Before vaccination, the infection of macaques with SARS-CoV-2 generated both anti-RBD antibodies (Fig. 2d) and low but detectable levels of antibodies inhibiting the binding of the spike protein to the ACE2 receptor (Fig. 2e). Before vaccination, low Th1 (IFN-γ$^{+/-}$ IL-2$^{+/-}$ TNF-α) type CD4$^+$ T-cell responses were observed for both groups of convalescent macaques following ex vivo stimulation of PBMCs with RBD and N-peptide pools (Fig. 2f; Supplementary Fig. 2e). None of the convalescent animals had detectable anti-RBD or anti-N CD8$^+$ T cells (Supplementary Fig. 2f).

Two weeks after αCD40.RBD vaccine injection, all six vaccinated macaques exhibited significantly increased levels of anti-S (Fig. 2b) and anti-RBD IgG (Fig. 2d) in the serum, which correlated with an increased capacity of inhibition of RBD binding to the ACE2 receptor ($p = 0.022$, Fig. 2e), as they remained elevated four weeks after vaccination. In an in vitro assay using authentic viruses[14], we confirmed that antibodies raised by the vaccine not only neutralizes the variant containing the D614G present in the αCD40.RBD (Supplementary Fig. 9), but also cross neutralizes B1.1.7 and to a lesser extent B1.351 known to be partially resistant to antibodies raised by previously circulating variants[31,32]. None of these parameters increased in PBS-injected convalescent controls (Fig. 2d, e; Supplementary Fig. 9a). In addition, anti-S IgG levels in the vaccinated macaques were higher ($p = 0.0018$) than those typically observed in humans 1 to 3 months after symptomatic SARS-CoV-2 infection (Supplementary Fig. 3c). The immunization also elicited a significant increase in the anti-RBD Th1 response in all six immunized animals ($p = 0.031$; Fig. 2f, g), whereas no changes in the magnitude of anti-N CD4$^+$ T cells (Supplementary Fig. 4e) or SARS-CoV-2 specific CD8$^+$ T cells was observed (Supplementary Fig. 4f).

**The αCD40.RBD vaccine improves the protection of convalescent macaques against SARS-CoV-2 reinfection**. Four weeks following vaccine or placebo injection, the 12 convalescent macaques were exposed a second time to a high dose ($1 \times 10^6$ pfu) of SARS-CoV-2 administered via the combined intra-nasal and intra-tracheal route using a previously reported challenge procedure[30]. Six SARS-CoV-2 naive animals were also challenged as controls.

All naive animals became infected, as shown by the detection of viral genomic (gRNA) and sub-genomic (sgRNA) RNA in tracheal (Fig. 3a–d; Supplementary Fig. 5a, b) and nasopharyngeal (Fig. 3d; Supplementary Fig. 5c–g) swabs and broncho-alveolar lavages (BAL, Fig. 3e and Supplementary Fig. 5h). Of note, the dynamics of viral replication in these animals was comparable to that observed during the first infection six months earlier in the two groups of convalescent macaques (Supplementary Fig. 3a, b). The non-vaccinated convalescent animals were not protected against the second SARS-CoV-2 challenge, but significantly lower viral RNA levels were detected in the upper respiratory tract than in the naive animals (Fig. 3a–e and Supplementary Fig. 5). The αCD40.RBD vaccine remarkably improved the partial protection observed in the convalescent macaques. All vaccinated animals exhibited significantly lower viral gRNA levels ($p = 0.015$, Fig. 3d) than the non-vaccinated convalescent animals. The levels of

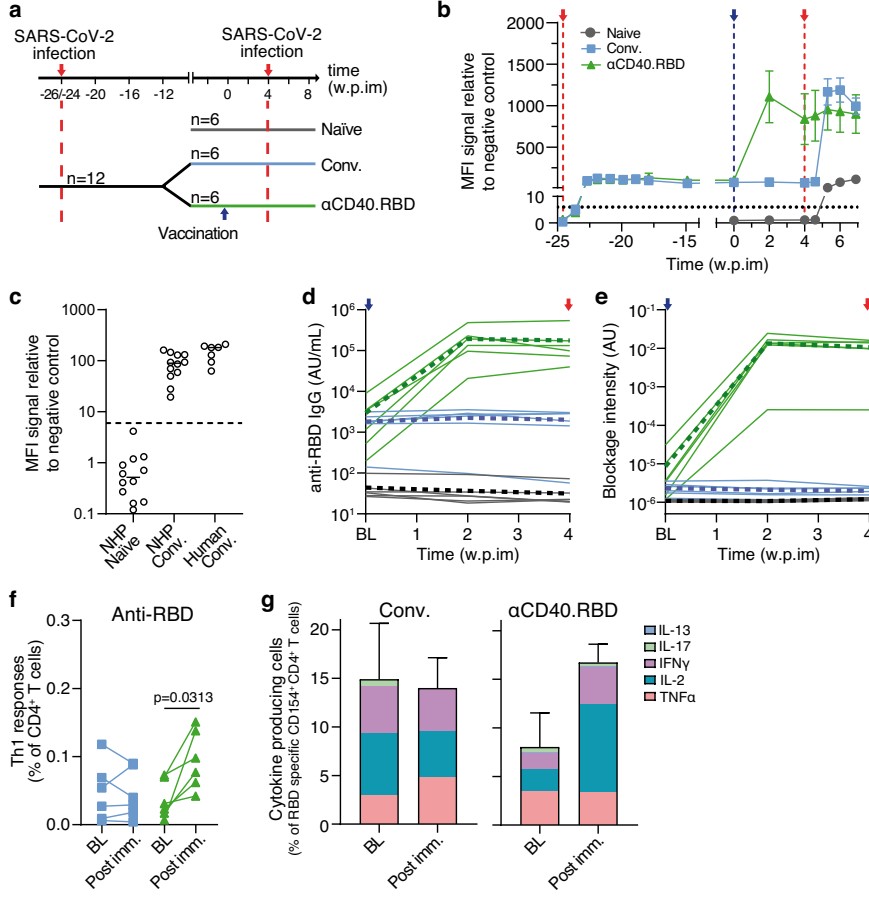

**Fig. 2 SARS-CoV-2 specific B- and T-cell responses induced by αCD40.RBD in convalescent NHP. a** Study design in cynomolgus macaques. **b** Relative MFI of IgG binding to SARS-CoV-2 S protein, measured using a Luminex-based serology assay, in serum samples (mean ± SD of 6 animals per group). The red and blue vertical dotted lines indicate viral exposure and vaccination, respectively. **c** SARS-CoV-2 S protein-specific binding before any exposure to SARS-CoV-2 (week −26) and on the week of vaccine injection (week 0) in macaques ($n = 12$) compared to convalescent humans ($n = 7$) sampled 24 weeks after the onset of symptoms. The horizontal dotted line represents the background threshold and bars indicate the mean of each group. **d** Quantification of SARS-CoV-2 antibodies against RBD measured in the serum of NHPs using a multiplexed solid-phase chemiluminescence assay. Each plain line indicates the individual values, and the bold dotted lines represent the mean for each experimental group. **e** Quantification of antibody-induced inhibition of ACE-2 binding in NHP serum. Symbols are as for **d**. **f** Frequency of RBD-specific Th1 CD4+ T cells (CD154+ and IFN-γ ± IL-2 ± TNF-α) in the total CD4+ T cell population for each non-immunized convalescent macaque ($n = 6$, blue lines and symbols) and αCD40.RBD-vaccinated convalescent macaque ($n = 6$, green lines and symbols). PBMC were stimulated overnight with SARS-CoV-2 RBD overlapping peptide pools. Time points in each experimental group were compared using the Wilcoxon signed rank test. **g** Frequency of cytokine producing cells in the RBD-specific CD4+ T cells (CD154+) for non-immunized convalescent macaque (left) and αCD40.RBD-vaccinated convalescent macaque (right). Each bar indicated the mean of the 6 vaccinated convalescent macaques ± SD. Distribution of cytokines is indicated within each bar. BL: Baseline approximately 1 week before immunization; "Post imm.": Two weeks post immunization.

sgRNA remained below the limit of detection in upper respiratory tract samples for 5 of 6 vaccinated animals, whereas sgRNA was detected in 4 of 6 non-vaccinated convalescent and all naive control animals (Supplementary Fig. 5b, g). Moreover, the time post-exposure (p.expo.) to reach undetectable gRNA levels was significantly lower in vaccinated convalescent than non-vaccinated and control animals (Fig. 3c and Supplementary Fig. 5e, log rank, $p < 0.0001$). The efficacy of vaccination was also higher in the lower respiratory tract, as only 3 of 6 vaccinated macaques were above the limit of detection for gRNA in BAL at day 3 p.expo. versus day 6 for the six non-vaccinated convalescent animals (Fig. 3e). Complete protection from shedding of the virus from the gastrointestinal tract was noted in the non-vaccinated and vaccinated convalescent macaques (Supplementary Fig. 5i), indicating that in addition to vaccine, the natural infection immunity could play an important role to prevent secondary viral transmission[33].

The reduction of viral load in vaccinated and non-vaccinated convalescent macaques relative to naive infected animals was associated with a limited impact on leukocyte numbers (Supplementary Fig. 6) and reduced cytokine concentrations in the plasma, in particular those of IL-1RA and CCL2 (Supplementary Fig. 7b). Such viral loads and cytokine profiles were also associated with a reduction in lung lesions (Fig. 3h and Supplementary Fig. 8), as scored by X-ray computerized tomography (CT).

We then analyzed the immune responses of all animals following SARS-CoV-2 viral challenge. The naive controls showed the slowest development of anti-S, anti-RBD, and anti-N IgG (Fig. 3f), of which the levels remained significantly lower than for the other two groups at day 20 p.expo. ($p = 0.022$). The non-vaccinated convalescent animals raised a rapid and robust anamnestic antibody response (Fig. 3f), which was associated with a significant increase ($p = 0.031$) in the serum capacity to

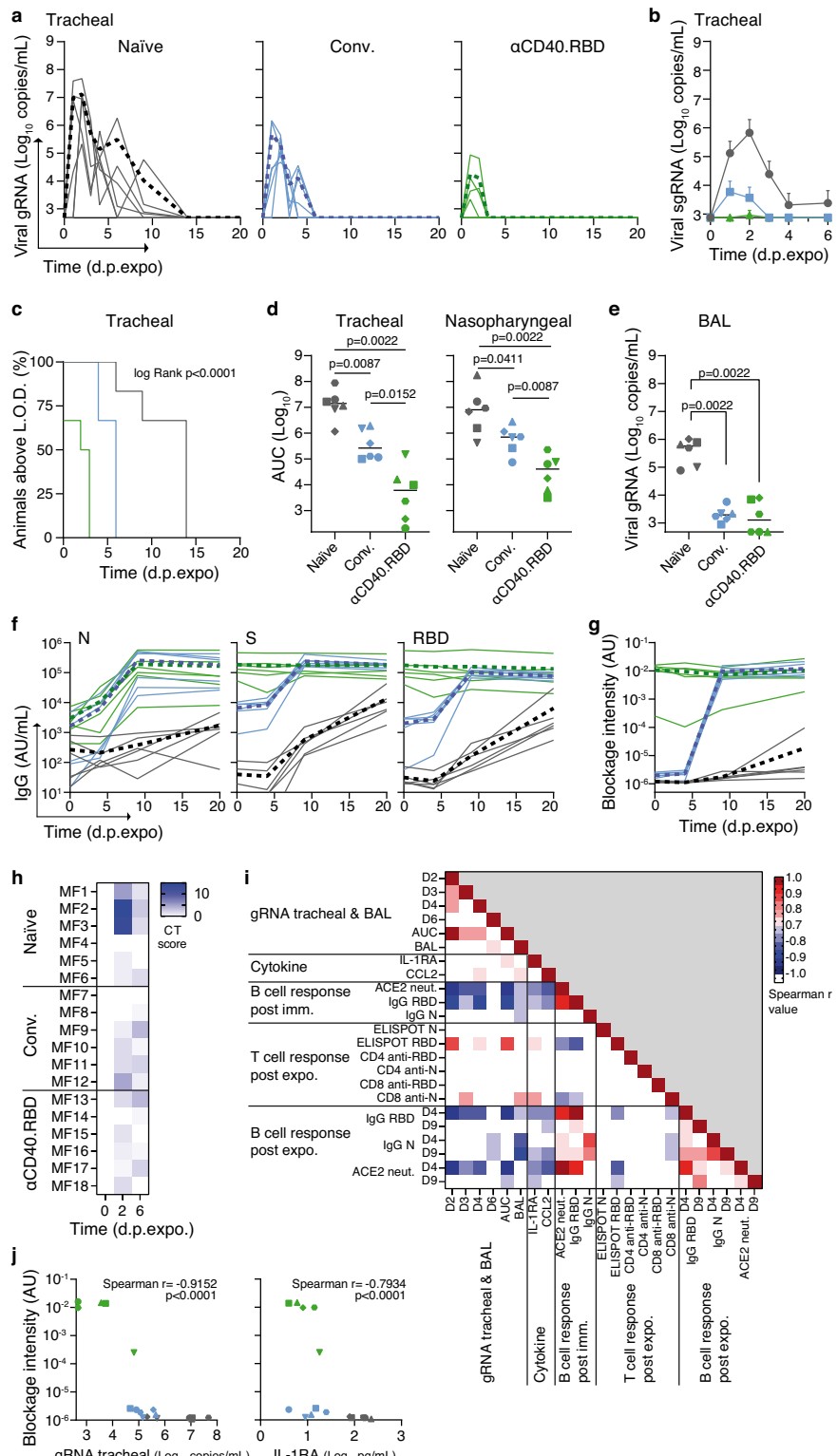

neutralize ACE2 binding to RBD (Fig. 3g) by p.expo. day 9, reaching at that time the levels observed in the vaccinated group. The anti-S- and anti-RBD-specific antibody responses and neutralization activity of the serum was maintained in the vaccinated macaques at the high levels already achieved at the time of challenge and remained superior to that of the control macaques (Fig. 3f, g). The anti-RBD Th1 CD4$^+$ response increased post-challenge for most of the control (convalescent

and naive) animals, with higher levels for some of the naive controls as early as p.expo. day 9 (Supplementary Fig. 4g). On the contrary, all 18 animals showed comparable antibody and CD4$^+$ T cell responses to the N-peptide pool (Supplementary Fig. 4g), probably reflecting a predominance of the response against non-structural antigens in infected individuals. The IFN-γ-mediated CD8$^+$ T-cell response was also mainly directed against the N peptides (Supplementary Fig. 4d), but with a significantly reduced

**Fig. 3 Efficacy of αCD40.RBD in convalescent cynomolgus macaques. a** Genomic viral RNA (gRNA) quantification in tracheal swabs of naive (left, gray lines), convalescent (middle, blue lines), and αCD40.RBD-vaccinated convalescent macaques (right, green lines). The bold line represents the mean viral load for each experimental group. **b** Mean of subgenomic (sgRNA) viral loads in tracheal swabs. Data are presented as mean values ± SD for each experimental group (n = 6 NHP/group). **c** Percentage of macaques with viral gRNA above the limit of detection (LOD) over time in tracheal swabs. Experimental groups were compared using log Rank tests; two-tailed p value is indicated. **d** Area under the curve (AUC) of gRNA viral loads in tracheal (left panel) and nasopharyngeal swabs (right panel). **e** gRNA viral quantification in BAL three days post-exposure (d.p.expo). **d**, **e** Each plot represents one macaque (n = 6 NHP/group) and bars indicate the mean of each group. Groups were compared using the two-tailed non-parametric Mann–Whitney test. **f** Quantification of SARS-CoV-2 IgG binding N, S, and RBD after challenge. Each plain line indicates individual values, and the bold dotted lines represent the mean for each experimental group. **g** Quantification of antibody-induced inhibition of ACE-2 binding. Lines as in **f**. **h** Lung CT-scores of macaques before challenge and at 2 and 6 d.p.expo to SARS-CoV-2. The CT score includes lesion type and lesion volume summed for each lobe. **i** Correlation matrix between virological and immune parameters. The heatmap indicates the Spearman r values (Only values between −0.7 and −1, and 0.7 and 1 are colored in the heatmap). **j** Correlation between antibody-induced inhibition of ACE-2 binding at 0 d.p.expo. and tracheal gRNA viral loads (left) or IL-1RA plasma concentration (right) at 2 d.p.expo. The Spearman r and p (two-tailed) values are indicated.

intensity in all convalescent macaques than in the naive controls (Supplementary Fig. 4h), probably reflecting the lower exposure to viral antigens as a result of better control of viral replication.

Spearman analysis between all recorded parameters revealed that the induction of anti-RBD- and ACE2-inhibiting antibodies was the strongest parameter to correlate with the reduction of viral load and disease markers, as were the plasma levels of the inflammatory cytokines IL-1RA and CCL2 (Fig. 3i and j).

## Discussion

In humans, the durability of protection induced by natural SARS-CoV-2 infection and the first vaccine candidates is unknown. In convalescent humans, the virus neutralizing-antibody response wanes and re-infections have been reported within months following previous exposure[33,34]. The decrease in neutralizing-antibody levels observed in most patients within three months post-infection may suggest that vaccine boosters will be required to provide long-lasting protection[35]. In contrast to previous NHP re-challenge studies performed shortly after a first infection[36], we demonstrate that SARS-CoV-2 reinfection is not fully prevented in convalescent macaques six months after initial exposure to the virus, confirming that protective immunity wanes over time. In addition, the vaccines currently used in humans are aimed at preventing severe disease and only partial information is available as to their capacity to prevent infection and reduce initial viral replication to the level required to significantly limit secondary transmission. Vaccinated individuals who develop an asymptomatic or mild symptomatic infection may continue transmitting the virus and actively contribute to circulation of the virus. The αCD40.RBD vaccine we developed significantly improved immunity of convalescent macaques, resulting in a reduction of viral load following re-exposure to the virus down to levels that may avoid such secondary transmission. This vaccine may therefore represent an appropriate booster of pre-existing immunity, either induced by natural infection or previous priming with vector-based vaccines. This new-generation subunit vaccine targeting the antigen to CD40-expressing cells, may have advantages for a safe and efficient boosting strategy. The capacity to induce protective immunity without requiring an adjuvant would accelerate the development of a protein-based vaccine with expected improved tolerability over adjuvanted vaccines and thus suitable for people with specific vulnerabilities and children, an important part of the population to consider in the control of circulation of the virus.

## Methods

**Ethics and biosafety statement animal studies**. The 20-week-old female NSG (NOD.Cg-Prkdcscid Il2rgtm1Wjl/SzJ) humanized mice (hu-mice) were supplied by the Jackson Laboratories (Bar Harbor, ME, USA) under MTA #1720. Five donors whose HLA typing is recapitulated in the supplemental table SI provided

hematopoietic stem cells for human immune system reconstitution of the mice. The level of human immune cells reconstitution reached an average of 70%. The hu-mice were housed in Mondor Institute of Biomedical Research infrastructure facilities (U955 INSERM-Paris East Creteil University, Ile-de-France, France) in micro-isolators under pathogen-free conditions with human care, at a temperature of 20–24 °C with 50% +/− 15% humidity and a 12-h light/12-h dark cycle. The protocols were approved by the institutional ethical committee "Comité d'Ethique Anses/ENVA/UPEC (CEEA-016)" under statement number 20-043 #25329. The study was authorized by the "Research, Innovation and Education Ministry" under registration number 25329-2020051119073072 v4.

Cynomolgus macaques (Macaca fascicularis), aged 37–58 months (8 females and 13 males) and originating from Mauritian AAALAC certified breeding centers were used in this study. All animals were housed in IDMIT facilities (CEA, Fontenay-aux-roses), under BSL-3 containment (Animal facility authorization #D92-032-02, Préfecture des Hauts de Seine, France) and in compliance with European Directive 2010/63/EU, the French regulations and the Standards for Human Care and Use of Laboratory Animals, of the Office for Laboratory Animal Welfare (OLAW, assurance number #A5826-01, US). The protocols were approved by the institutional ethical committee "Comité d'Ethique en Expérimentation Animale du Commissariat à l'Energie Atomique et aux Energies Alternatives" (CEtEA #44) under statement number A20-011. The study was authorized by the "Research, Innovation and Education Ministry" under registration number APAFIS#24434-2020030216532863v1.

**Vaccination of humanized mice**. The hu-mice received immunizations at week 0, 3, and 5. The priming injection was an intraperitoneal administration of 10 μg of αCD40-RDB adjuvanted with 50 μg of polyinosinic-polycytidylic acid (Poly-IC; Invivogen) combined or not with an intramuscular injection of DREP-S (10 μg). Then hu-mice received booster i.p injections of αCD40-RDB (10 μg) plus Poly-IC (50 μg). Blood was collected at weeks 0 (before immunization), 3, and 6. Hu-mice were euthanized at week 6.

**Non-human primate study design**. Convalescent cynomolgus macaques previously exposed to SARS-CoV-2 and used to assess hydroxychloroquine (HCQ) and azithromycin (AZTH) antiviral efficacy. None of the AZTH neither HCQ nor the combination of HCQ and AZTH showed a significant effect on viral replication[5]. Six months (24–26 weeks) post infection (p.i.), twelve of these animals were randomly assigned in two experimental groups. The convalescent vaccinated group (n = 6) received 200 μg of αCD40.RBD vaccine by subcutaneous (SC) route diluted in PBS and without any adjuvant. The other six convalescent animals were used as controls and received the equivalent volume of PBS by SC. The two groups of convalescent animals were sampled at week 2 and 4 following vaccine or PBS injection for anti-SARS-CoV-2 immune response evaluation. Additional six age matched (43.7 months ±6.76) cynomolgus macaques from same origin were included in the study as controls naïve from any exposure to SARS-CoV-2.

**Evaluation of anti-Spike, anti-RBD, and IgG inhibiting antibodies**. Anti-Spike IgG from human and NHP sera were titrated by multiplex bead assay. Briefly, Luminex beads were coupled to the Spike protein as previously described[6] and added to a Bio-Plex plate (BioRad). Beads were washed with PBS 0.05% tween using a magnetic plate washer (MAG2x program) and incubated for 1 h with serial diluted individual serum. Beads were then washed and anti-NHP IgG-PE secondary antibody (Southern Biotech, clone SB108a) was added at a 1:500 dilution for 45 min at room temperature. After washing, beads were resuspended in a reading buffer 5 min under agitation (800 rpm) on the plate shaker then read directly on a Luminex Bioplex 200 plate reader (Biorad). Average MFI from the baseline samples were used as reference value for the negative control. The amount of anti-Spike IgG was reported as the MFI signal divided by the mean signal for the negative controls. Human sera from convalescent patients who were hospitalized

with virologically confirmed COVID-19 were collected three months after symptoms recovery and used as controls for the titration of anti-Spike antibodies.

Anti-RBD and anti-Nucleocapside (N) IgG were titrated using a commercially available multiplexed immunoassay developed by Mesoscale Discovery (MSD, Rockville, MD) as previously described[7]. Briefly, antigens were spotted at 200 – 400 µg/mL in a proprietary buffer, washed, dried, and packaged for further use (MSD® Coronavirus Plate 2). Then, plates were blocked with MSD Blocker A following which reference standard, controls and samples diluted 1:500 and 1:5000 in diluent buffer were added. After incubation, detection antibody was added (MSD SULFO-TAG$^{TM}$ Anti-Human IgG Antibody) and then MSD GOLD$_{TM}$ Read Buffer B was added and plates read using a MESO QuickPlex SQ 120MM Reader. Results were expressed as arbitrary unit (AU)/mL.

The MSD pseudo-neutralization assay was used to measure antibodies neutralizing the binding of the spike protein to the ACE2 receptor. Plates were blocked and washed as above, assay calibrator (COVID-19 neutralizing antibody; monoclonal antibody against S protein; 200 µg/mL), control sera, and test sera samples diluted 1:10 and 1:100 in assay diluent were added to the plates. Following incubation of the plates, an 0.25 µg/mL solution of MSD SULFO-TAG$^{TM}$ conjugated ACE-2 was added after which plates were read as above. Electrochemioluminescence (ECL) signal was recorded and results expressed as 1/ECL.

**Experimental infection of macaques with SARS-CoV-2.** Four weeks after immunization, all animals were exposed to a total dose of $10^6$ pfu of SARS-CoV-2 virus (hCoV-19/France/ lDF0372/2020 strain; GISAID EpiCoV platform under accession number EPI_ISL_406596) via the combination of intranasal and intratracheal routes (0.25 mL in each nostril and 4.5 mL in the trachea, i.e. a total of 5 mL; day 0), using atropine (0.04 mg/kg) for pre-medication and ketamine (5 mg/kg) with medetomidine (0.05 mg/kg) for anesthesia. Nasopharyngeal, tracheal and rectal swabs, were collected at 1, 2, 3, 4, 6, 9, 14, and 20 days post exposure (d.p.exp.) while blood was taken at 2, 4, 6, 9, 14, and 20 d.p.exp. Bronchoalveolar lavages (BAL) were performed using 50 mL sterile saline at 3 d.p.exp in order to be close to the peak of viral replication and to be able to observe a difference between the vaccinated and control groups. In our earlier study[30], we found that at later time-points, viral loads in the BAL were very low or negative. Chest CT was performed at baseline and at 2 and 6 d.p.exp. on anesthetized animals using tiletamine (4 mg/kg) and zolazepam (4 mg/kg). Lesions were scored as we previously described[30]. Blood cell counts, hemoglobin, and hematocrit were determined from EDTA blood using a DXH800 analyzer (Beckman Coulter).

**Statistical analysis.** Data were collected using classical Excel files (Microsoft Excel 2016). Differences between unmatched groups were compared using an unpaired t-test or the Mann–Whitney U test (Graphpad Prism 8.0), and differences between matched groups were compared using a paired t-test or the Wilcoxon signed-rank test (Graphpad Prism 8.0). Viral kinetic parameter was compared using log-rank tests (Graphpad Prism 8.0). Correlation between viral and immune parameter was determined using nonparametric Spearman correlation (Graphpad Prism 8.0).

**Reporting summary.** Further information on research design is available in the Nature Research Reporting Summary linked to this article.

## Data availability

Data that support the findings of this study are provided in the source data file (NHP experiments and uncropped gel picture) of this paper and are available from the corresponding author upon reasonable request. Source data are provided with this paper.

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

## Acknowledgements

We thank B. Delache, S. Langlois, J. Demilly, N. Dhooge, P. Le Calvez, M. Potier, J. M. Robert, T. Prot, and C. Dodan for the NHP experiments; L. Bossevot, M. Leonec, L. Moenne-Loccoz, M. Calpin-Lebreau, and J. Morin for the RT-qPCR, ELISpot and Luminex assays, and for the preparation of reagents; W. Gros for NHP T-cell assays and flow cytometry; B. Fert for her help with the CT scans; S. Tricot for fluorescent labeling of Ab used in flow cytometry; M. Barendji, J. Dinh and E. Guyon for the NHP sample processing; S. Keyser for the transports organization; F. Ducancel and Y. Gorin for their help with the logistics and safety management; I. Mangeot for here help with resources management and B. Targat contributed to data management; We thank S. Bellil and V. Enouf for contribution to viral stock challenge production and A. Nougairede for sharing the plasmid used for the sgRNA assays standardization. The mice picture in Fig. 1 was created with BioRender.com. The programme was funded by the Vaccine Research Institute via the ANR-10-LABX-77 grant. Studies in hu-mice model have been supported by ARN grant ANR-20-COV6-0004-01. The Infectious Disease Models and Innovative Therapies (IDMIT) research infrastructure is supported by the "Programme Investissements d'Avenir", managed by the ANR under reference ANR-11-INBS-0008. The Fondation Bettencourt Schueller and the Region Ile-de-France contributed to the implementation of IDMIT's facilities and imaging technologies. The NHP study received financial support from REACTing, the Fondation pour la Recherche Medicale (FRM; AM-CoV-Path) and the European Infrastructure TRANSVAC2 (730964) for implementation of in vivo imaging technologies an NHP immuno assays. The virus stock used in NHPs was obtained through the EVAg platform (https://www.european-virus-archive.com/), funded by H2020 (653316).

## Author contributions

Y.L., G.Z. and R.L.G. Conceptualization, study design, supervision, data analysis, project administration, and writing. R.M., M.G., V.G., and S.C: Methodology, validation, formal analysis, data curation, figures elaboration Writing—Original Draft, Visualization; R.M. and M.C: project administration. S.Z., Z.W., Y.L., J.E.: Design and production of αCD4.RBD vaccines. I.S. and P.L.: Design and production of DREP-S vaccine. N.D.B.: development of viral load assays and analysis, T cell assays project administration. A.S.G., M.G.P.: Supervision, T cell analysis, and flow cytometry. S.C., C.F., G.P., M.C.: serology assays in NHP and analysis. P.M. and V.C.: contributed to supervising NHP studies. J.L., F.R., R.H.T.F., L.D.: animal studies. T.N., N.K., C.C.: in vivo imaging and CT analysis.

S.V.D.W.: viral stock production and characterization, data analysis. M.P., R.T.: data analysis; D.P., T.B., O.S.: measurement and analysis of the neutralizing activity of the sera against infectious SARS-CoV-2 variants. A.W., C.L., and M.S.: contributed to the development of immune assays.

## Competing interests

The funders had no role in study design, data collection, data analysis, data interpretation, or data reporting. The authors S.Z., G.Z., V.G., M.C and Y.L, are named inventors on patent applications based on this work held by Inserm Transfert. The remaining authors declare no competing interests.
