## [Peer Review File · Nature Communications]

Targeting SARS-CoV-2 receptor-binding domain to cells expressing CD40 improves protection to infection in convalescent macaquesReviewers' Comments:

Reviewer #1:

Remarks to the Author:

The manuscript of Marlin R et al. is a follow up of previous studies from the group looking at the potential of targeting diverse viral antigens to CD40 expressing antigen-presenting cells as an efficient way to elicit strong antigen-specific T- and B-cell responses, but in this case applied to the development of a new generation of subunit vaccines against SARS-CoV-2. The authors demonstrate that targeting the receptor-binding domain (RBD) of the SARS-CoV-2 spike protein to CD40 (α CD40.RBD) induces specific B and T cells in a humanized mouse model. Moreover, a single dose of the α CD40.RBD vaccine administered in convalescent non-human primates (NHPs) improved protection against a new high-dose virulent challenge. The study is interesting and will be of significance to the SARS-CoV-2 vaccine field but some of the data are not supported.

1. Based on the results from Fig. 1E, 1I and Extended data Fig. 2, the authors argued that α CD40.RBD efficiently boosted (DREP)-S primed B- and T-cell SARS-CoV-2 specific responses. However the data does not support this conclusion.

In terms of B cells: At three weeks after the priming injection, when the animals only received 1 dose of the immunogens, the SARS-CoV-2 S protein-specific IgG-switched human B-cell frequencies within the hu-B cells in the blood of hu-mice was higher in Gr3 ((DREP)-S+ α CD40.RBD) than in Gr2 (α CD40.RBD) (Fig. 1E). However, at six weeks after the priming injection (when the 3 doses were completed), the frequencies of SARS-CoV-2 S protein-specific IgG-switched human B-cell within the hu-B cells in the spleen of hu-mice were two-folds lower in Gr3 ((DREP)-S+ α CD40.RBD/ α CD40.RBD/ α CD40.RBD) vs Gr2 (α CD40.RBD/ α CD40.RBD/ α CD40.RBD) (Fig. 1I). Similarly, as shown in extended data Fig2A, the expression of CXCR4 and CCR10 in the B-specific cells were lower in Gr3 vs Gr2. In view of these findings why the authors consider that α CD40.RBD vaccine efficiently boosted (DREP)-S primed B specific response?

In terms of T cells: In extended data Fig.2B-C the authors show the frequency of antigen-specific Th1 human CD4+ T cells (IFN γ +/- IL2+/- TNF α) and the antigen-specific human CD8+ T cells (IFN γ +) in the total human CD4+ T-cell or human CD8+ T-cell population, respectively, for non-immunized (mock) and vaccinated hu-mice at 6 weeks post injection.

-Why are there only 4 individual data represented in the vaccinated group (Gr2) instead of 10?

-What is the T cell specific responses in group 3 ((DREP)-S+ α CD40.RBD/ α CD40.RBD/ α CD40.RBD)?

The authors should include the T cell response of this group in order to support that α CD40.RBD vaccine efficiently boosted (DREP)-S primed T specific response.

-What are the percentages of hu CD8 T cell population in a naïve hu-mice?

-Which are the levels of IL2 and TNF α in the antigen-specific hu CD8 T cell population?

- At week 6 (1 week after the last boost) probably most of the specific T cells induced in hu-mice have an effector phenotype. A better characterization including memory specific markers should be used in order to better define the phenotype of these cells.

2. It will be helpful to include in Fig 2(G) the frequency of cytokine producing cells in the RBD-specific CD4+ T cells (CD154+) for non-vaccinated convalescent macaques.

3. As the authors point in the result section (Fig. 3F and 3G) the anti-S- and anti-RBD-specific antibody responses and neutralization activity of the serum was maintained in the vaccinated macaques at the high levels already achieved at the time of challenge and remained superior to that of the control macaques BUT during the first 10 days after virus exposure . At day 10 post exposure the non-vaccinated convalescent macaques reached the same antibody levels that the vaccinated animals.

4. As shown in extended data Fig. 5I the complete protection from shedding of the virus from the gastrointestinal tract was noted NOT ONLY in the immunized macaques, BUT ALSO in the non-immunized convalescent macaques indicating that in addition to vaccine, the natural infection immunity could play an important role to prevent secondary viral transmission.

Minor point:

-Update in the introduction the licensed vaccines against the SARS-CoV2.

Reviewer #2:

Remarks to the Author:

In this manuscript, the authors performed two animal experiments including NSG HIS mice and convalescent NHPs to demonstrate that immunization with anti-CD40 fused RBD protein could activate human and macaque antigen-presenting cells and induced RBD-specific B cell, Th1-biased CD4+ response and CD8+ response. Ex vivo stimulation of the splenocytes demonstrate the functional phenotype of the T cells responses as well. Immunization with the anti-CD40 RBD protein in convalescent NHP protected the animals from a secondary infection represented by reduced viral gRNA copies in the respiratory tracts of the animals and lower lung CT score.

Major concerns:

-An important control group is missing for the mice and NHP studies, in which one group of animals should be immunized with just the RBD without the anti-CD40 antibody. It's understandable that NHP is not easily available, but this control group should be included at least in the mouse study. The advantage of anti-CD40 RBD over just the RBD is not demonstrated. For example, no serological data nor protection data is shown to suggest anti-CD40 RBD is more immunogenic, protective or induce more long-lasting immunity than the RBD itself.

-A vaccine-induced recall response could be important to prevent a secondary infection of the same virus or infection of other variants. It has been shown that spike/RBD specific and NP-specific memory B cells were generated in COVID-19 convalescent patients. It would not be a surprise that the anti-CD40 RBD induced recall responses in convalescent NHPs. But due to the lack of the RBD only control group, the advantage of the anti-CD40 RBD is not well demonstrated.

-The modeling did not seem to add much value to the study as the decreased of the viral load in the upper respiratory tract would suggest the reduced shedding of the virus, therefore, limiting the transmission

Minor concerns:

-Figure 1D, since group 2 and 3 are adjuvanted with polyIC, the author should include this information in the figure panel.

-The author should measure serum IgG titer/neutralizing antibody induced by indicated regimen in mice. It wasn't clear how immunogenic the protein is in naïve animals.

-Extended Data figure 2A, did the authors intend to show RBD specific memory B cells responses here?

RESPONSES TO REVIEWERS' COMMENT

Reviewer #1 (Remarks to the Author)

The manuscript of Marlin R et al. is a follow up of previous studies from the group looking at the potential of targeting diverse viral antigens to CD40 expressing antigen-presenting cells as an efficient way to elicit strong antigen-specific T- and B-cell responses, but in this case applied to the development of a new generation of subunit vaccines against SARS-CoV-2. The authors demonstrate that targeting the receptor-binding domain (RBD) of the SARS-CoV-2 spike protein to CD40 (α CD40.RBD) induces specific B and T cells in a humanized mouse model. Moreover, a single dose of the α CD40.RBD vaccine administered in convalescent non-human primates (NHPs) improved protection against a new high-dose virulent challenge. The study is interesting and will be of significance to the SARS-CoV-2 vaccine field but some of the data are not supported.

1.Hu-mice: *Based on the results from Fig. 1E, 1I and Extended data Fig. 2, the authors argued that α CD40.RBD efficiently boosted (DREP)-S primed B- and T-cell SARS-CoV-2 specific responses. However the data does not support this conclusion. In terms of B cells: At three weeks after the priming injection, when the animals only received 1 dose of the immunogens, the SARS-CoV-2 S protein-specific IgG-switched human B-cell frequencies within the hu-B cells in the blood of hu-mice was higher in Gr3 ((DREP)-S+ α CD40.RBD) than in Gr2 (α CD40.RBD) (Fig. 1E). However, at six weeks after the priming injection (when the 3 doses were completed), the frequencies of SARS-CoV-2 S protein-specific IgG-switched human B-cell within the hu-B cells in the spleen of hu-mice were two-folds lower in Gr3 ((DREP)-S+ α CD40.RBD/ α CD40.RBD/ α CD40.RBD) vs Gr2 (α CD40.RBD/ α CD40.RBD/ α CD40.RBD) (Fig. 1I). Similarly, as shown in extended data Fig2A, the expression of CXCR4 and CCR10 in the B-specific cells were lower in Gr3 vs Gr2. In view of these findings why the authors consider that α CD40.RBD vaccine efficiently boosted (DREP)-S primed B specific response?*

Reply: We do agree with the reviewer that this point needs to be clarified. In the two vaccine groups, the prime boost strategy containing α CD40.RBD efficiently elicited B- and T-cell SARS-CoV-2 specific responses. The frequency of spike specific B cells is equivalent after

three injections of the α CD40.RBD vaccine alone (Group 2) or after two booster injections with the α CD40.RBD following a priming in combination with the (DREP)-S vaccine (Group 3). We may thus surmise that combined use of α CD40.RBD with vector based vaccines do not affect the quality of the spike specific humoral response. This may allow a more flexible use of the α CD40.RBD vaccine, alone or in combined heterologous prime-boost strategies. The sentence of conclusion of this paragraph was tempered accordingly (page 7 lines 163-168).

2. Hu-mice: *In terms of T cells: In extended data Fig.2B-C the authors show the frequency of antigen-specific Th1 human CD4⁺ T cells (IFN γ ⁺/- IL2⁺/- TNF α) and the antigen-specific human CD8⁺ T cells (IFN γ ⁺) in the total human CD4⁺ T-cell or human CD8⁺ T-cell population, respectively, for non-immunized (mock) and vaccinated hu-mice at 6 weeks post injection. Why are there only 4 individual data represented in the vaccinated group (Gr2) instead of 10?*

Reply: In Extended data Fig.2B-C, we only used splenocytes from hu-mice reconstituted with hematopoietic stem cells of HLA-A*0201 or HLA-A*0301 haplotype donors in order to monitor both, the secretion of cytokines by T cells, and the induction of HLA-A*0201 or HLA-A*0301 tetramer⁺ SARS-CoV-2 RBD-protein-specific human CD8⁺ T cells. Besides, we performed these *in vitro* assays using thawed splenocytes and had to pool cells from two or three HLA-A*0201 or HLA-A*0301 hu-mice to get enough cells. This explains the difference in numbers between the previous tests and this *in vitro* assay. This is now clarified in the figure's legend.

3.Hu-mice: *What is the T cell specific responses in group 3 ((DREP)-S+ α CD40.RBD/ α CD40.RBD/ α CD40.RBD)? The authors should include the T cell response of this group in order to support that α CD40.RBD vaccine efficiently boosted (DREP)-S primed T specific response.*

Reply: In the initial Extended Fig.2B-C, we presented experiences performed in mock animals and vaccinees comprising animals from both Gr2 and Gr3 since the objective was mainly to demonstrate that α CD40.RBD containing regimen elicited RBD-specific CD4⁺ exhibiting a Th1 profile, IFN- γ ⁺ CD8⁺ T cells and Tetramer⁺ CD8⁺ T cells but not to compare vaccine groups.

However, to better answer to reviewer's comment, we have performed new experiments in a cohort of HLA-A*0201 hu-mice vaccinated to complete the data with four animals in each Gr2 and Gr 3. Spleen cells were dedicated to *in vitro* re-stimulation assay. We propose to add these new results in the revised manuscript (new Extended data Fig.2 D-G and below) showing an induction of functional specific CD4⁺ and CD8⁺ T cells in vaccinated hu-mice from both groups with no significant difference between both vaccination regimens.

New Extended data Fig. 2D-E. (D) Frequency of antigen-specific Th1 human CD4⁺ T cells (IFN γ +/- IL2+/- TNF α) and (E) antigen-specific human CD8⁺ T cells (IFN γ +) in the total human CD4⁺ T-cell or human CD8⁺ T-cell population, respectively, for non-immunized (mock) and vaccinated hu-mice. (F) Representative flow-cytometry plots showing the HLA-A*0301 and HLA-A*0201 tetramer+ SARS-CoV-2 RBD-protein-specific human CD8⁺ T cells for poly(IC)-injected hu-mice and α CD40.RBD-vaccinated hu-mice (group 2). (G) Frequency of RBD-specific human CD8⁺ T cells (Tetramer+) in the total human T cell population for non-immunized (mock) and vaccinated hu-mice.

4.Hu-mice: What are the percentages of hu CD8 T cell population in a naïve hu-mice?

Reply: human CD8⁺ T cells account for about 26% (25.9% \pm 4%, mean \pm SD) and 10% (9.8% \pm 4.5%, mean \pm SD) of human immune cells in the spleen and blood of naïve NSG hu-mice, respectively. These counts are likely to fluctuate with the age of hu-mice. We show below (for the reviewers only), the blood CD8⁺ T cell frequencies in our preimmunization cohort of NSG hu-mice which were in fact comparable between groups.

Frequency of CD8⁺ T cells in the blood of our NSG hu-mice before their immunization with the α CD40.RBD vaccine or control injections (PBS or Poly:IC). Empty circle: PBS group; Full circle: poly:IC group; square: α CD40.RBD homologous prime/boost; Diamond: α CD40.RBD heterologous prime/boost.

5.Hu-mice: *Which are the levels of IL2 and TNF α in the antigen-specific hu CD8 T cell population?*

Reply: the frequencies of IL-2⁺ antigen-specific hu-CD8⁺ T cells were 0% [0-0.72] (median, IQR; 25%-75%) for Gr2 and 0.02% [0-0.06] for Gr3, with no statistical difference between the two groups. The frequencies of TNF⁺ cells were 1.48% [0.79-5] for Gr2 and 0.83% [0.46-2.35] for Gr3, again without any statistical difference between the two groups. This is now reported in the legend of the Extended data Fig.2

6.Hu-mice: *At week 6 (1 week after the last boost) probably most of the specific T cells induced in hu-mice have an effector phenotype. A better characterization including memory specific markers should be used in order to better define the phenotype of these cells.*

Reply: to better answer to the referee and in order to evaluate the pool of memory T cells, we have added new data the frequency of effector memory T cells in the blood of hu-mouse collected at 3 and 6 week post-immunization (w.p.im). These results are now shown in extended data in the new Extended Fig 2B-2C of the revised version. We confirmed that the α CD40.RBD vaccine used either in homologous or heterologous vaccination induced effector

memory CD4⁺ and CD8⁺ T cells at both time points, both regimens of vaccination being equally effective.

Extended data Fig 2B-2C (in the revised version of the manuscript): Induction of effector memory CD4⁺ and CD8⁺ T cells in hu-mouse blood by the vaccination regimens. Results are expressed as fold change at 3 and 6 w.p.im compared to baseline values. Empty circle: PBS group; Full circle: poly:IC group; square: α CD40.RBD homologous prime/boost (group 2); Diamond: α CD40.RBD heterologous prime/boost (group 3).

7.NHP: It will be helpful to include in Fig 2(G) the frequency of cytokine producing cells in the RBD-specific CD4⁺ T cells (CD154⁺) for non-vaccinated convalescent macaques.

Reply: The requested data is now included in Fig 2G.

8.NHP: As the authors point in the result section (Fig. 3F and 3G) the anti-S- and anti-RBD-specific antibody responses and neutralization activity of the serum was maintained in the vaccinated macaques at the high levels already achieved at the time of challenge and remained superior to that of the control macaques BUT during the first 10 days after virus exposure. At day 10 post exposure the non-vaccinated convalescent macaques reached the same antibody levels that the vaccinated animals.

Reply: The reviewer's comment is correct indeed. We have now commented this point in the main text, page 9, lines 250-251.

9. NHP: As shown in extended data Fig. 5I the complete protection from shedding of the virus from the gastrointestinal tract was noted NOT ONLY in the immunized macaques, BUT ALSO in the non-immunized convalescent macaques indicating that in addition to vaccine, the

natural infection immunity could play an important role to prevent secondary viral transmission.

Reply: The reviewer's comment is correct indeed. We have now commented this point in the main text, page 9, lines 236-238.

Reviewer #1 minor point:

10. Update in the introduction the licensed vaccines against the SARS-CoV2.

Reply: Introduction was updated as requested, page 4, lines 71-79.

Reviewer #2 (Remarks to the Author):

In this manuscript, the authors performed two animal experiments including NSG HIS mice and convalescent NHPs to demonstrate that immunization with anti-CD40 fused RBD protein could activate human and macaque antigen-presenting cells and induced RBD-specific B cell, Th1-biased CD4⁺ response and CD8⁺ response. Ex vivo stimulation of the splenocytes demonstrate the functional phenotype of the T cells responses as well. Immunization with the anti-CD40 RBD protein in convalescent NHP protected the animals from a secondary infection represented by reduced viral gRNA copies in the respiratory tracts of the animals and lower lung CT score.

Major concerns:

1. NHP & hu-mice: An important control group is missing for the mice and NHP studies, in which one group of animals should be immunized with just the RBD without the anti-CD40 antibody. It's understandable that NHP is not easily available, but this control group should be included at least in the mouse study. The advantage of anti-CD40 RBD over just the RBD is not demonstrated. For example, no serological data nor protection data is shown to suggest anti-CD40 RBD is more immunogenic, protective or induce more long-lasting immunity than the RBD itself. A vaccine-induced recall response could be important to prevent a secondary infection of the same virus or infection of other variants. It has been shown that spike/RBD specific and NP-specific memory B cells were generated in COVID-19

convalescent patients. It would not be a surprise that the anti-CD40 RBD induced recall responses in convalescent NHPs. But due to the lack of the RBD only control group, the advantage of the anti-CD40 RBD is not well demonstrated.

Reply: Our group has already published several papers testing different DC targeting vaccines showing the advantages of a targeted protein as compared to the protein alone with different adjuvant. The hu-mice and NHP studies we report in the manuscript is an extension to SARS-CoV-2 of use of the anti-CD40 vaccine platform we developed for other type of microbial antigens, mainly for influenza and HIV vaccines. We privileged the targeting of CD40 among many other receptors expressed by antigen presenting cells (APC), and dendritic cells (DC) in particular. These studies are presented in references 8-18-21-23-24-27 of the main text.

In particular, we reported in *ex vivo* assays using human donor cells that influenza M1 antigen, used at doses as low as 10 pM and delivered via anti-CD40, elicited robust Flu M1-specific T cell responses. At least 1000-fold higher doses of Flu M1 alone or isotype control hIgG4.FluM1 were required to elicit detectable Ag-specific responses (Flamar et al, 2012; doi:10.4049/jimmunol.1102390).

In the hu-mice model, we have recently published (reference 18 of the manuscript) the comparison between α CD40.HIV.Env (CD/CD) *versus* IgG4.HIV.Env (non-targeted protein, IgG4/IgG4) (figure below).

Extended Data Fig. 4: Evaluation of specific humoral immune responses elicited by the IgG4-Env control construct. Frequency of Env-specific IgG⁺ hu-B cells at week 6 in the blood of hu-mice immunized with the IgG4-Env plus CpG, the α CD40.Env vaccine (CD/CD) or control hu-mice injected with CpG (n=3) or PBS (n=2).

Individual values are presented, along with the median. Two-sided Mann-Whitney U-test was used for the comparison. * $p < 0.05$. (V. Godot et al, *Plos pathogen*, *PLoS pathogens* 16, e1009025 (2020)).

In non-human primates, we have reported that targeting HA influenza antigen to DC receptors (DC-ASGPR and Lox-1) influences the quality of the antigen specific T-cell response in macaques (Li D et al, 2012; doi: 10.1084/jem.20110399). In addition, we demonstrated that the targeting of Gag HIV antigen to DC through anti-Langerin antibodies significantly improves in macaques the Gag specific antibody response, as compared to Gag protein alone or associated to control IgG4, without the need to use an adjuvant (Epaulard O et al, 2014, doi: 10.4049/jimmunol.1303339).

Moreover, we recently generated data demonstrating that targeting HIV envelope gp140 to CD40 using the same antibody backbone for the α CD40.RBD vaccine induced significantly higher specific T cell and antibody response in cynomolgus macaques than the same Env antigen fused to the non-targeting IgG4 control. This data is now presented in Extended data Fig. 10 and commented in the main text page 5, line 113-115 and page 7, line 174-175.

Extended Data Fig 10. Comparison of immunogenicity of α CD40.HIV_{env} vaccine to the non-targeting isotype control IgG4.HIV_{env} in cynomolgus macaques. (A) Study design including three injections of the vaccines (200 μ g by subcutaneous route) co-injected with poly-IC-LC (Iltanol, 1 mg by subcutaneous route). (B) HIV-Env specific response as measured by IFN γ ELISPOT following overnight stimulation of NHP PBMC with a pool of 15-mer overlapping peptides spanning the Env gp140. Area under the curve (AUC) from day 0 to day 50 of the immunization. (C) Anti- HIV-Env IgG in serum of immunized macaques. Area under the curve from day 0 to day 50 of the immunization. Each plot represents one macaque. The mean of 6 NHPs is indicated by horizontal line. Groups were compared using the non-parametric Mann-Whitney test.

2. NHP: A vaccine-induced recall response could be important to prevent a secondary infection of the same virus or infection of other variants. It has been shown that spike/RBD specific and NP-specific memory B cells were generated in COVID-19 convalescent patients. It would not be a surprise that the anti-CD40 RBD induced recall responses in convalescent NHPs. But due to the lack of the RBD only control group, the advantage of the anti-CD40 RBD is not well demonstrated.

Reply: This comment is partially overlapping the previous request for adding a group of immunized animals with not targeted RBD. We try to enrich our manuscript with data obtained from macaques immunized against HIV through CD40 targeting or with not targeted antigen. To address the first point of this reviewer question, we included new data demonstrating that the antibodies generated against RBD in the group of NHP vaccinated with α CD40.RBD also neutralize in a replicating virus assay the new circulating variants, and in particular B1.351.. The latter is known to have reduced susceptibility to cross-neutralization by antibodies against previously circulating SARS-CoV-2 strains. This is now reported in Extended Fig.9 and commented in the main text page 8, lines 200-1204. The three scientists who performed these assays have been added to the list of the authors.

3. Biostats: The modeling did not seem to add much value to the study as the decreased of the viral load in the upper respiratory tract would suggest the reduced shedding of the virus, therefore, limiting the transmission.

Reply: We tend to agree with the reviewer's and considered that improving the information that modeling can bring to this manuscript would require significantly extended developments. If the Editor's agree, we suggest to remove this part from the revised manuscript.

Minor concerns:

4.Hu-mice: Figure 1D, since group 2 and 3 are adjuvanted with polyIC, the author should include this information in the figure panel.

Reply: we would like to thank the reviewer for helping us to clarify this point. The information has been added in the revised figure panel.

Fig1D. Schematic overview of vaccination strategies in NSG humanized (hu) mice, including three experimental groups, 9 to 10 animals/group.

5.Hu-mice: The author should measure serum IgG titer/neutralizing antibody induced by indicated regimen in mice. It wasn't clear how immunogenic the protein is in naïve animals.

Reply: We and others have shown that measurements of specific IgG in hu-mice is very difficult since the final maturation of B cells in this model is not optimal and only very low levels of IgG are found following immunization. Usually, as previously published by our group and others, the evaluation of specific B-cell responses is better assessed by the frequency of circulating or spleen specific B cells. Moreover, the production of detectable IgG levels is highly dependent of the hu-mice model. For example, in a previous study, we reported that the CD40 targeting vaccination in the presence of CpG-B as an adjuvant induced a more robust B-cell response in NRG hu-mice (V. Godot et al, Plos pathogen, PLoS pathogens 16, e1009025 (2020)) with an average level of circulating IgG around 14,9 microg/mL (median, see graph below) which was enough to assess the specificity of these IgG. In the present paper, we vaccinated NSG hu-mice, the only hu-mice available during the locked-down of research laboratories in spring 2020 when the study was initiating. It is well

known that the NSG hu-mice produce very low levels of circulating IgG. We confirmed this fact by comparing the total blood IgG levels between the NRG and NSG hu-mice vaccinated with our α CD40 vaccines (see the figure below). The NSG hu-mice produced 3 to 14 time less hu-IgG than the NRG ones depending on the vaccination regimen, which was too low to determine their specificity or their neutralizing capacities. Thus, using a luminex assay we were not able to detect anti-RBD IgG in vaccinated mice.

Comparison of circulating IgG levels in NSG (in bleu) and NRG (in pink) hu-mice after α CD40 targeting vaccination (for reviewer only)

6.Hu-mice Extended Data figure 2A, did the authors intend to show RBD specific memory B cells responses here?

Reply: we indeed intended here to show RBD-specific memory B cells responses by manual back-gating of SARS-CoV-2 S protein-specific IgG-switched human B cells onto hCXCR4 versus hCD19 and hCCR10 versus hCD19 dot blots. We have now realized, through the reviewer's comment, that the legend of Extended Fig 2A is incomplete and therefore does not assist in its understanding. We added in the revised version of the manuscript that bulk human B cells were colored in grey and S-specific IgG⁺ human B cells were colored in red (see below).

Extended Fig2A. Manual backgating of SARS-CoV-2 S protein-specific IgG-switched human B cells onto hCXCR4 versus hCD19 and hCCR10 versus hCD19 dot blots prepared from concatenated data from all hu-mice in each group. Bulk human B cells were colored in grey and S-specific IgG⁺ human B cells were colored in red (new legend Extended data Fig 2)

Reviewers' Comments:

Reviewer #1:

Remarks to the Author:

The authors have addressed all the concerns in the revision. The paper could be published in the present format.

Reviewer #3:

Remarks to the Author:

The authors sufficiently addressed most of this reviewer's comments, except adding the RBD control group for the mouse study. The authors justified the experiment by showing a similar study comparing anti-CD40 fused HIV Env and IgG4 fused HIV Env. To some extent, the HIV study showed proof of principle of targeting CD40-expressing APC. In the opinion of this reviewer, there could be difference in constructing Env and RBD. For example, was there a trimerization domain in the anti-CD40-Env, that could target APC better with 3 copies of the anti-CD40? The same amount of a much larger anti-CD40-RBD would also have less RBD epitopes than those in RBD alone; Would the fused RBD has a different conformation than RBD alone, etc. It'll be of interest to show the benefit of anti-CD40-RBD over RBD. A minor point, it'll be helpful if the authors could show a schematic illustration of their anti-CD40 RBD construct.

Point by point response to the reviewers:

We thank Reviewer #1 who considers that, “The authors have addressed all the concerns in the revision. The paper could be published in the present format”,

We also thank Reviewer #3 who addressed several comments that certainly will improve our manuscript. We hope that our responses will be favorably consider by the reviewer.

1. **The reviewer commented that,** *“The authors justified the experiment by showing a similar study comparing anti-CD40 fused HIV Env and IgG4 fused HIV Env. To some extent, the HIV study showed proof of principle of targeting CD40-expressing APC. In the opinion of this reviewer, there could be difference in constructing Env and RBD. For example, was there a trimerization domain in the anti-CD40-Env, that could target APC better with 3 copies of the anti-CD40?”*

We agree that the HIV Env and RBD constructs may not have fully comparable immunogenic properties. However, the two anti-CD40 vaccines have been constructed using identical strategies to anchor the viral antigens. Both vaccines have one monomeric HIV Env or RBD fused to the each heavy chain forming the Fc domain of the IgG4. In total each construct have two monomers of Env or RBD similarly attached to the anti-CD40 platform. The reviewer may certainly refer to different HIV vaccine candidates using a stabilized trimeric form the Env glycoprotein, which is not the case in the α CD40.HIV we used in our proof of concept study in mice and macaques. We therefore surmise that both constructs should have similar capabilities in targeting the antigen presenting cells expressing human CD40. The effect on CD40 expressing cells of the α CD40.RBD vaccine is also confirmed in the figure 1 of our manuscript.

We are now providing this information **in lines 166 to 174 of the main text**, as indicated in “track-changes” format.

As per request of the Reviewer #3, we are also providing in **Fig.1 and Extended Data Fig10** pictures of both RBD and HIV Env anti-CD40 vaccines.

2. **The reviewer also suggest that** *“The same amount of a much larger anti-CD40-RBD would also have less RBD epitopes than those in RBD alone”.*

Authors response: Regarding the α CD40.RBD construct, the alignment of amino acids (aa) and epitopes after fusion to the C-terminal part of the Fc region of the heavy chain, allows the conservation of all epitopic regions of the RBD sequence. One could anticipate that only a few (or any) aa located near the fusion protein would lose their antigenic capacity, which is not likely. Moreover our epitope mapping studies predict that this upstream region of RBD do not contain major epitopes for anti-SARS-CoV-2 neutralizing antibodies. Finally, the hypothesis raised by the reviewer would mean a low antigenicity of the anti-CD40 construct. The results we obtained in humanized mice and in non-human primates demonstrate that the α CD40.RBD have good capacities to induce antibodies neutralizing virus infectivity, including for new P1 and B 1.351 variants.

Reviewers' Comments:

Reviewer #3:

Remarks to the Author:

The authors addressed the comments sufficiently